# An Online Algorithm for Routing an Unmanned Aerial Vehicle for Road Network Exploration Operations after Disasters under Different Refueling Strategies

Lorena Reyes-Rubiano [1,2,*,†] , Jana Voegl [3,†] and Patrick Hirsch [3,*,†]

1   Operations Management, Otto-von-Guericke University Magdeburg, D-39016 Magdeburg, Germany
2   International School of Economic and Administrative Sciences, Universidad de La Sabana,
    Bogotá 53753, Colombia
3   Institute of Production and Logistics, University of Natural Resources and Life Sciences Vienna,
    331180 Vienna, Austria; jana.voegl@boku.ac.at
*   Correspondence: lorena.reyes@ovgu.de (L.R.-R.); patrick.hirsch@boku.ac.at (P.H.);
    Tel.: +43-1-47654-73419 (P.H.)
†   These authors contributed equally to this work.

**Abstract:** This paper is dedicated to studying on-line routing decisions for exploring a disrupted road network in the context of humanitarian logistics using an unmanned aerial vehicle (UAV) with flying range limitations. The exploration aims to extract accurate information for assessing damage to infrastructure and road accessibility of victim locations in the aftermath of a disaster. We propose an algorithm to conduct routing decisions involving the aerial and road network simultaneously, assuming that no information about the state of the road network is available in the beginning. Our solution approach uses different strategies to deal with the detected disruptions and refueling decisions during the exploration process. The strategies differ mainly regarding where and when the UAV is refueled. We analyze the interplay of the type and level of disruption of the network with the number of possible refueling stations and the refueling strategy chosen. The aim is to find the best combination of the number of refueling stations and refueling strategy for different settings of the network type and disruption level.

**Keywords:** disrupted road network; immediate response operations; on-line algorithm; refueling strategies; online exploration strategies; network cutting procedure; labeled network

## 1. Introduction

In emergencies, humanitarian logistics plans and coordinates resources for providing relief to vulnerable people in regions affected by a disaster [1]. The main objective of humanitarian logistics is to mitigate the negative impacts of a disaster and manage a fast response and recovery to the emergency. Response and recovery operations depend on an initial assessment of the affected area. This paper focuses on the assessment operation, which is executed during or immediately after the disaster, depending on the type of disaster [2]. The objective of this operation is to determine the state of the infrastructure and the needs of the affected population. Thus, response and recovery operations depend on this assessment operation.

Recently, unmanned aerial vehicles (UAVs), or aircraft that operate without a human onboard, have attracted significant attention in humanitarian logistics. UAV use offers an increment of the speed and flexibility of humanitarian operations [3]. Based on the study of Reyes-Rubiano et al. [4], we study the evaluation operation using a UAV with flying range limitations. Reyes-Rubiano et al. [4] focus on the evaluation of the state of the road network. The main objective of this operation is to determine the roads that can be used for the distribution of humanitarian aid or evacuation of the affected population. We study this operation as an on-line UAV routing problem. The routing decisions involve the road

network and the aerial network of the affected area. Once a disaster occurs, some roads cannot be used because they have been affected by a natural, man-made, or technological hazard [5].

In current practice, organizations have been starting to include UAVs in response and reconstruction operations [6]. The use of UAVs promises cost efficiency for assessing destroyed road networks after a disaster. A UAV has a flying range that limits the number of affected zones that can be explored. Flying range limitations add a real risk of route failure because the UAV runs out of fuel mid-route [6–10]. UAV routing decisions in response operations are inefficient and do not meet the needs of humanitarian logistics due to uncertainty. As a result, several initiatives are aiming to develop tools to improve the reliability of on the fly routing decisions and to improve the efficiency of response operations [9].

The problem tackled in this paper is to assess the state of a post-disaster area using a UAV with flying range limitations. Based on the principle that most humanitarian relief operations are conducted using the road network, the main objective of our study is to determine the road network that is still functioning after a disaster. We consider that the only information available to start the exploration is the state of the road network in a pre-disaster situation and the location of the sites where victims are located. We refer to the road network in a pre-disaster situation as the known road network. The post-disaster road network is denoted as the disrupted network. We deal with uncertainty regarding the state of the road network by making routing decisions on the fly. The UAV starts at the Disaster Management Center (DMC); here, refueling is always possible. As the UAV advances its exploration, partial information regarding the state of the disrupted network is extracted, and new routing decisions are made. The UAV flies over the road network with a camera to transfer in real time a video to the DMC to determine if the explored edge is a functional or disrupted road. The objective of this exploration is to assess the road network state and evaluate the accessibility to the location of the victims, i.e., villages. The exploration is conducted with a UAV with a limited flying range due to the capacity of the fuel tank. During the exploration, functional roads and facilities detected determine the disrupted network. Refueling stations at the victim locations can be used to refuel the UAV and avoid a route failure. We assume that fuel stations that can be resupplied by road can be used as refueling stations during the exploration. Thus, refueling stations can be supplied if a functional path from the DMC to the refueling station exists. Thus, at the beginning of the exploration, it is unknown which refueling stations could be used during the exploration. Motivated by this problem, we intend to answer the following research question:

What are suitable strategies for refueling a UAV with a limited flying range under the uncertainty of the status of a disrupted network?

To answer this research question, we use different networks, which are described next:

**Known road network**
The known road network refers to the road network of a rural region in pre-disaster conditions (see Figure 1). All locations in the region are connected by the road network, and there is at least one path between each pair of locations. The set of locations in the known road network are the DMC, road crossings, and victim locations. In this study, the DMC is the center of logistics operations, from where disaster information, fuel, and other resources are managed and humanitarian aid is deployed. Therefore, the DMC must be connected to the victim nodes by functional roads; otherwise, the victims are unreachable using the road network, and it makes no sense to conduct an exploration or plan a deployment.

**Disrupted network**
Functional parts of the known road network (see Figure 2). This network contains the same locations as the known road network. In the disrupted network, refueling stations at victim locations can be used for refueling the UAV, if they are connected to the DMC by functional roads.

**Aerial network**

The aerial network is a fully connected network. As shown in Figure 3, it contains all the locations of the known road and the disrupted network. All locations are connected to each other, including the disruption locations. The number of nodes and arcs in the network increases each time a disruption is detected. In this aerial network, the locations of the disruptions are included as disrupted locations. Thus, the number of nodes and edges in the aerial network increases along with the progress of the exploration.

We extend the algorithm presented in the study Reyes-Rubiano et al. [4]. We propose three refueling strategies and different numbers of possible refueling station in the disrupted network. Our solution approach aims to balance the cost of considering a safety fuel buffer and the cost of performing a refueling operation every time it is needed. We extend the algorithm of Reyes-Rubiano et al. [4] by adding the following procedures: (1) Detection of refueling stations reachable by road from the DMC. This procedure determines which refueling stations are operating at each time $t$ of the exploration. (2) Dynamic safety buffer. This procedure calculates the dynamic safety buffer in terms of distance. The dynamic buffer represents the distance that the UAV has to fly to explore the next selected unexplored road edge and then return to the closest reachable refueling station to the current UAV location. The safety buffer depends on the current location of the UAV and the closest refueling station, using an aerial edge. (3) A procedure that controls the UAV flying capacity. Depending on the refueling strategy defined in advance, the current location of the UAV, flying capacity, and the road network information, the algorithm makes two decisions: (a) it decides whether or not to perform a refueling operation, and (b) it decides at which station the refueling operation is most convenient. The refueling strategies under the different numbers of potential refueling stations are compared in extensive numerical studies using instances from the literature with different sizes and various disruption levels in the road network. Thus, the paper extends the existing literature as follows:

- This paper extends the online algorithm of Reyes-Rubiano et al. [4] to conduct routing decisions to explore a road network whose status and available resources are initially unknown.
- By routing decisions in a network that is updated dynamically during the exploration using a UAV with a limited flying range.
- This study presents refueling strategies based on a dynamic fuel buffer that avoids the risk of having a route failure due to running out of fuel.
- We propose three criteria for refueling decisions based on the information available from the road network and the flying capacity at each moment of the exploration.
- Our research considers that as long as a refueling station is detected, which is accessible from the DMC, it can be used to refuel the UAV. Road accessibility is evaluated to ensure that the station can be resupplied with fuel.
- Our research assesses the state of the road network, including refueling stations, and maps the disrupted network to use in the humanitarian aid deployment operation.
- Our research provides information about which percentage of potential refueling stations is best for different network sizes and disruption levels, and thus supports tactical decision-making.
- We additionally provide information on the best refueling strategy when the number of potential refueling stations is fixed, i.e., when a disaster hits for a certain network size and disruption level.

The rest of this paper is structured as follows. In Section 2, a literature review is presented. Section 3 provides a formal description of the problem addressed. The solution approach is presented in Section 4. The numerical study and computational results are provided in Sections 5 and 6. Finally, Section 7 presents the main conclusions and proposes future work.

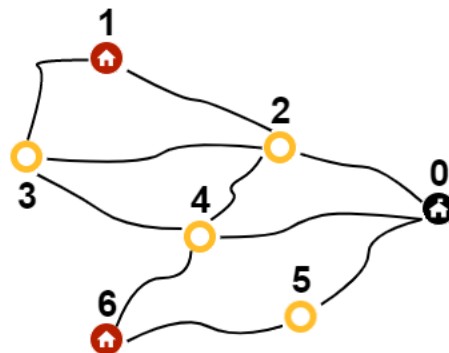

**○ DMC   ⊙ Victims   ○ Road crossings**

**‿ Road edges**

**Figure 1.** Known road network.

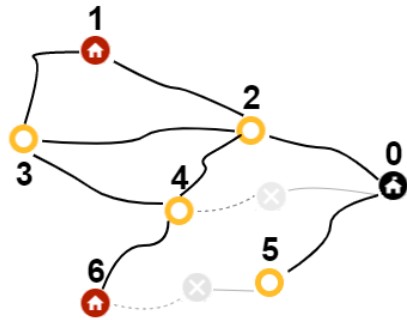

**○ DMC   ⊙ Victims   ○ Road crossings**

**‿ Road edges   ⊗ Disruption locations**

**⋯ Disrupted road edges**

**Figure 2.** Disrupted network.

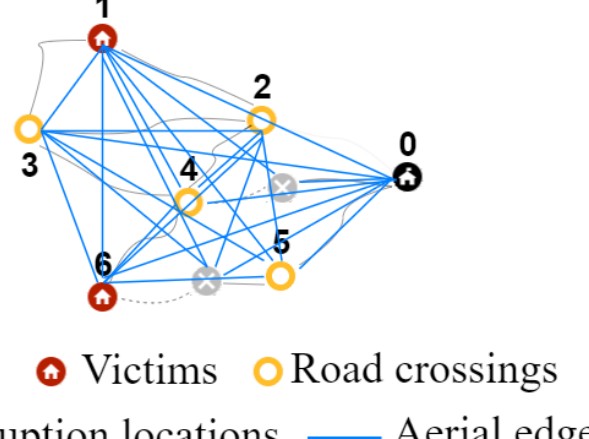

**○ DMC   ⊙ Victims   ○ Road crossings**

**⊗ Disruption locations   —— Aerial edges**

**Figure 3.** Aerial network.

## 2. Literature Review

This section is dedicated to the use of UAVs in humanitarian logistics operations. Reyes-Rubiano et al. [4] present a consolidation of work dealing with disrupted networks in the field of humanitarian logistics. The relevance of the use of UAVs in disaster situations is important to achieve a rapid management of the situation. The reason is that UAVs have abilities to perform some difficult or dangerous tasks, providing high mobility and safety, as well as low cost [11]. In response, the number of studies focused on the integration of UAVs into logistics and information systems has grown. Rejeb et al. [11] present a review of works using UAVs and vehicles. One of the main fields of study focuses on technical limitations such as payload capacity and flying range of UAVs. This section focuses on the strategies that allow dealing with flying range limitations considering the maximum amount of fuel that a UAV can carry in the tank.

### 2.1. Fuel Management Strategies of UAVs

UAVs can use an electric motor, internal combustion engine, or hybrid engine. The advantage of electric UAVs is the flexibility in terms of size and weight: they are small and light aerial vehicles. Electric UAVs can operate with a replaceable or rechargeable battery. However, their flying range does not exceed 50 km [12], and the recharge time of the batteries is very long. UAVs with an internal combustion engine are much heavier, but the ratio between weight and flying range presents a good performance [13]. Another advantage of internal combustion UAVs is the fast speed of the refueling operation [12,13].

One of the strategies to deal with the flying range limitation is to force the inclusion of intermediate stops within the UAV route to perform refueling operations [14]. Another strategy is to perform battery assignment and scheduling for the UAV based on the time between charges and discharge time [15]. In the case of electric UAVs, battery management involves battery charging and storage management, whereby recharging stations demand a specialized type of infrastructure [16]. Boukoberine et al. [12] present a literature review focusing on strategies to deal with the technical limitations of UAVs.

The refueling strategies and routing decisions reported in the literature involve refueling station design. Planning routes under collaborative environments is another strategy to deal with flying range limitations. Zhang et al. [17] present an approach in which a fleet of UAVs operates in collaboration with a fleet of trucks. The trucks operate as refueling stations. The truck and drone routes are defined in advance, where intermediate stops to recharge the drone battery are included in the routing problem. Alvarez et al. [18] tackle the problem of determining the location of drone facilities. The problem is approached as a p-median, where one wants to maximize the coverage of the stations by locating stations that can quickly respond to the needs of UAVs. The authors recognize that the number of stations and their location depend on the points that demand travel routes and the network structure. The decision regarding the number of stations and where to locate them is oriented toward minimizing the travel distance. The authors propose a heuristic approach to solve a deterministic problem dealing with unmanned aerial vehicle station location.

Dezan et al. [19] propose a Bayesian network to predict the environment in which the drone will perform, based on a set of possible situations, which allows a failure mode and effects analysis. This information is useful for the estimation of speed and fuel consumption. Most studies focus on predicting fuel consumption and are based on stochastic travel times to determine route reliability [17]. The contribution of our research is to develop a systematic method that humanitarian relief agencies and decision-makers can use to determine which type of refueling strategy is more convenient for their specific region.

### 2.2. Humanitarian Operations and Disrupted Road Networks

This subsection focuses on literature related to response operation in a disrupted road network. Disruptions on a road network caused by a disaster deteriorate the accessibility of affected areas [20]. Some studies relate the concept of disrupted networks to situations

where there are shortages and accessibility issues of essential commodities such as water and food [21]. Loree and Aros-Vera [22] recognize that the deployment of relief operations depends on an assessment process immediately after the disaster. The authors focus on determining the optimal location of distribution centers to minimize the social cost. The authors propose a model that considers the deprivation cost to quantify the negative impact of not reaching and providing rapid relief to victim locations. Rath and Gutjahr, as well as Nolz et al. [21,23] aim to locate additional warehouses in the affected zone to deal with disruptions associated with shortages of relief supplies. They focus on determining the location and capacity of warehouses to deal with the scarcity of relief supplies. Hatefi et al. and Elçi and Noyan et al. [24,25] develop a stochastic programming model with several probable scenarios for determining the expected disrupted road network after an earthquake. Similarly, Reza et al. [26] deal with a disrupted network in which demand, travel time, travel distance, loading–unloading time, and costs are unknown. The authors propose a solution method to solve the problem of locating a limited number of relief centers and the problem of the distribution of perishable products based on expected values. Nezhadroshan et al. [27] propose a stochastic programming model to deal with the uncertainty in the supply chain. The authors consider that other repetitions of the primary disaster may occur and worsen the humanitarian aid distribution network. The authors solve a facility location problem and a product distribution problem considering the presence of multiple disaster events. In addition, the authors consider the cost of shortages or excess inventories.

Other studies focus on solving the problem of road accessibility to victims for the provision of services such as medical assistance and evacuation. The authors affirm that disrupted road networks induce inequality problems due to the limited accessibility of victims. Kunz and Wassenhove [28] study the vehicle fleet required to reach the affected areas and maximizing the coverage of humanitarian assistance to all victim locations using the road network. Duque and Sörensen [29] develop a model to plan the recuperation process of roads in an affected area. The authors focus on the repair of roads that improve the accessibility to victim locations. Koch et al. [30] develop a study in which a network disrupted by either traffic or disaster is simulated. The objective of this study is to determine which roads are the least likely to be disrupted in order to include them in the pathways for the provision of medical services. In this way, the authors intend to mitigate the negative impact of network disputes. Noyan et al. [31] develop a two-stage model considering different types of decision-makers. The authors address the problem of the distribution of humanitarian aid aimed at minimizing the inequality of resources that reach the victims. Hatefi and Jolai [32] propose a robust optimization model to minimize the operational cost and to define the functional road network after a disaster. The authors also consider disruptions related to the shortage of relief supplies at the DMC. Shahparvari et al. [33] develop a MIP model and genetic algorithm to determine the number of vehicles needed and travel routes to use to evacuate a disaster-affected area. The goal is to determine the most reliable set of routes given a risk of encountering a fire-disrupted street. The authors calculate all paths between pairs of nodes and given the risk to determine which route is the most reliable. Safitri and Chikaraishi [5] recognize the vulnerability and importance of the road network in emergency situations. The authors focus on studying the impacts of disruptions on the performance of the transportation network and trip demand to provide humanitarian assistance. The study concludes that highly disrupted networks require less travel time due to disruptions disconnecting the DMC and the vehicle being unable to move forward on the road network.

Table 1 presents a summary of the literature related to humanitarian operations that rely on road networks that can be disrupted. Most works address the problem of unknown information in a disrupted road network. They assume that disruptions and uncertainty can be modeled by stochastic approaches. In a post-disaster situation or during a disaster, the state of the area and the needs of the population are unknown [2,34]. Motivated by the problem of disrupted networks in the deployment of humanitarian aid, our first research

focuses on determining exploration strategies that lead to assessing the accessibility of the victims [4].

This paper extends our previous research by considering flying range limitations and including three different refueling strategies, as well as different numbers of possible refueling stations.

**Table 1.** Summary of the literature related to humanitarian operations and disrupted road networks.

| Authors | (a) | (b) | (c) | (d) | Objective |
|---|---|---|---|---|---|
| [5] | x | | | x | Measure impact of a disaster |
| [30] | x | | x | | Maximize reliability of distribution routes |
| [20] | x | | x | | (i) Maximize coverage<br>(ii) Maximize equity of relief supply |
| [21] | x | | | x | (i) Minmax. tour length from depot to victims<br>(ii) Maximize coverage of victims |
| [22] | x | | | x | Minimize social cost |
| [23] | x | | | x | (i) Minimize transport costs<br>(ii) Minimize fixed costs for depots and vehicles<br>(iii) Maximize coverage |
| [24] | | | x | | Minimize operational costs |
| [25] | x | | x | | (i) Minimize risk of a disaster<br>(ii) Minimize total shipping costs of relief supplies |
| [26] | x | | | x | (i) Minimize distance traveled<br>(ii) Minimize cost of opening and operating a relief center<br>(iii) Minmax. tour length from depot to victims<br>(iv) Minimize total quantity of perished items |
| [27] | x | | x | | (i) Minimize expected total cost<br>(ii) Minimize cost variability<br>(iii) Minimize customer satisfaction |
| [29] | x | | | x | Minimize weighted sum of shortest paths from depot to victims |
| [31] | x | | x | | Maximize equity of relief supply |
| [32] | x | | x | | Minimize expected total cost |
| [33] | x | | | x | Maximize number of people evacuated |
| **This research** | | x | | x | Minimize time to assess road accessibility to victims |

(a) complete information; (b) on-line information; (c) stochastic information; (d) deterministic information

## 3. Problem Description

The tackled problem consists of determining exploration and refueling strategies to evaluate the accessibility of victims by road using a UAV. The evaluation of the known road network is addressed as an on-line UAV routing problem with flying range limitations, aimed at minimizing the evaluation time to determine the accessibility of all victim locations by road.

The problem is described as a network $G = (N, E)$, where $N$ is the set of nodes referring to road-crossings, victim nodes, and the DMC. $E$ is the set of functional edges that represent functional roads in the pre-disaster situation. $\hat{G}_t = (\hat{N}_t, \hat{E}_t)$ represents the disrupted road network. $\hat{E}_t$ refers to the set of functional edges and non-explored edges at time $t$ of exploration. $\hat{N}_t$ is the set of nodes associated with road-crossings, victim nodes, the DMC, and disrupted nodes. Now, let $\hat{AG}_t = (N_t, A_t)$ be the fully connected aerial network where $A_t$ represents the set of edges connecting the set of nodes $N_t$ in the aerial network at time $t$. The number of nodes and edges of the aerial network grows as disruptions are detected. The location of the disruption on the road becomes a disrupted node connected to the aerial network. The routing problem involves both the disrupted network and the aerial network simultaneously (see Figure 4).

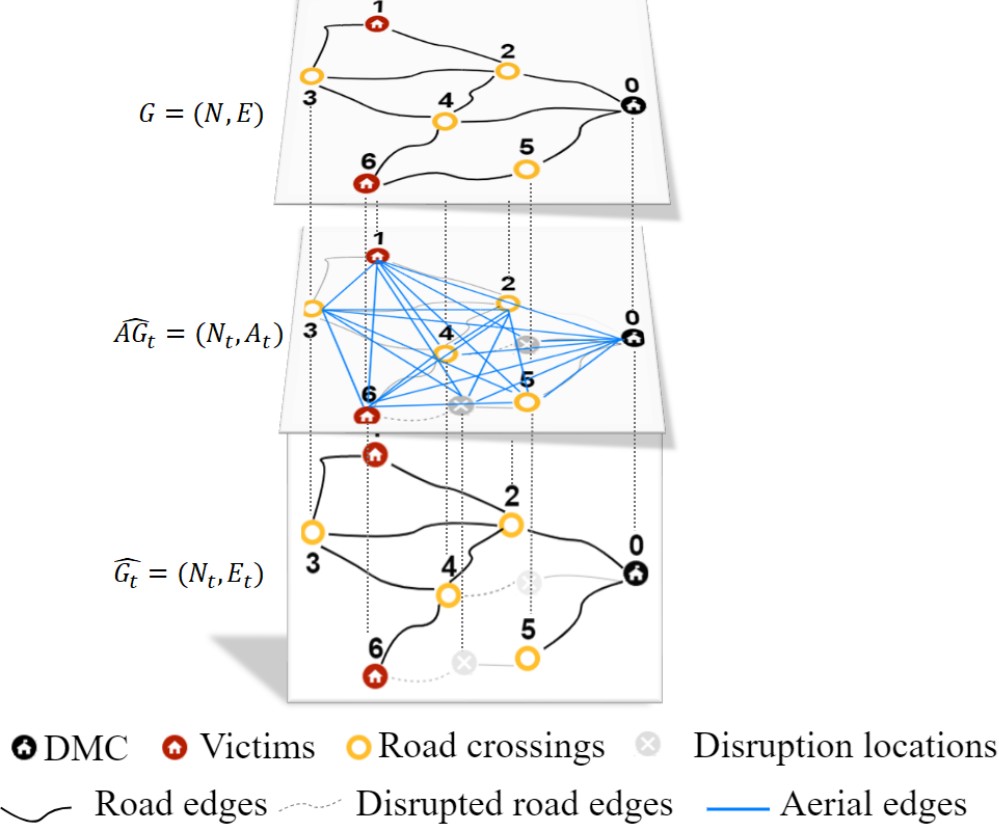

**Figure 4.** Example of the relationship between networks involved in the on-line UAV routing problem.

The problem of on-line routing with limited flying range involves the risk of having a route failure because the UAV runs out of fuel. We assume that the UAV selected to perform the exploration has a minimum flying range that allows it to go and return from the DMC to the farthest node of the network. We define the flying range in terms of distance. An example of a UAV used for exploration of small and medium instances is the Super Bat UAV with a flying range of 643 km and a maximum flying speed of 83 km/h [35]. For large instances, the U-Max UAV with a flying range of 1852 km and a maximum flying speed of 160 km/h is suitable [35]. We assume a speed of 60 km/h for the UAVs to allow for a good exploration of the road network.

Figures 5 and 6 show the difference a refueling station at a victim location can make. Figure 5 shows the situation with only the DMC is working as a refueling station, while in Figure 6, victim node 1 is a potential refueling station. This refueling station can be used in case victim location 1 is connected to the DMC by functional roads. In Figures 5 and 6, the UAV starts at the DMC. It first explores the road to road crossing 5 (0,5). Next, the UAV tries to reach victim location 6 using (5,6). However, a disruption is detected, and the UAV returns to the DMC directly using an aerial edge. Starting again at the DMC, the UAV explores the edges (0,2) and (2,1). It has now reached victim node 1. It now wants to explore the edge (1,3). However, there is not enough fuel left to reach road crossing 3 and return to the DMC. Figure 5 shows that the UAV returns directly to the DMC on an aerial edge due to the fuel shortage. In Figure 6, there is a different solution to the fuel shortage. Victim location 1 is a potential refueling stations. It is already established that there are functional roads (0,2) and (2,1) connecting it to the DMC. Thus, the UAV can refuel at victim node 1 and then continue by exploring (1,3).

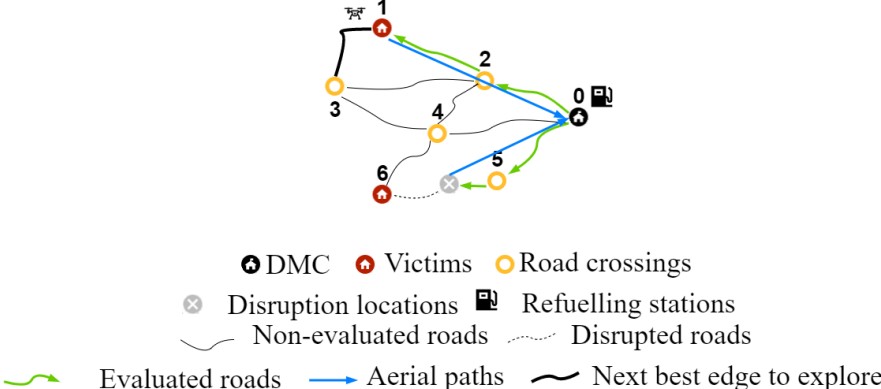

**Figure 5.** Example of the flying range limitation with only the DMC working as a refueling station.

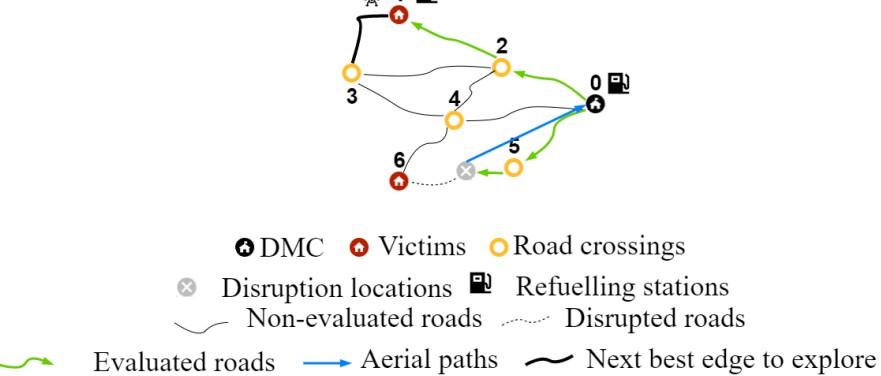

**Figure 6.** Example of the flying range limitation with the DMC and victim location 1 working as a refueling station.

For the tackled problem, we made the following assumptions:

1.  The road and the aerial network are undirected networks.
2.  Both the road and the aerial network are used: if an interruption is detected on a road, the route of the UAV is redirected using the aerial network.
3.  The UAV route starts and ends at the DMC.
4.  In the pre-disaster stage, all roads of the known network work. No initial information about the road network state after the disaster is given. The exploration of the road network starts under this assumption.
5.  The length of each edge on the road and the aerial network is known.
6.  The disruptions are not known in advance and might occur on any edge of the road network after the disaster.
7.  We assume a constant speed of the UAV.
8.  A pilot on the ground monitors the flight progress at the DMC.
9.  The pilot determines if a road is disrupted or not.
10. Based on the extracted real-time information, it is determined if a victim location is reachable by road or not.
11. The DMC and a certain percentage of the victim locations can operate as refueling stations. Victim locations that can operate as refueling stations must be connected to the DMC by road.
12. Staff to refuel the UAV are available at refueling stations.
13. The exploration duration is defined as the flying time and the refueling time of the UAV.
14. The type of UAV to use depends on the characteristics of the known road network to be explored. The flying range of the UAV should be at least the distance of the longest aerial edge connecting the DMC to the farthest node in the network.

Assumptions 11 to 14 extend Reyes-Rubiano et al. [4].

## 4. On-line Algorithm

This section presents the extension of the algorithm in [4]. We consider a conservative exploration strategy that integrates an orientation strategy. The conservative exploration strategy to deal with disruptions is a back and forward movement strategy. This strategy redirects the UAV based on the connected edges and nodes that have been identified up to the moment the interruption takes place. An aerial edge is used by the UAV to return to the closest connected node with unexplored adjacent edges. The orientation strategy determines how the priorities of edges are defined, while the movement strategy changes the exploration route of the UAV when a disruption is detected. The on-line algorithm considers one orientation criteria used to assign a priority to each edge that needs to be explored. We consider a weighted orientation criterion that depends on the edge connectivity and the edge length. The priority value of each edge is the weighted sum of both criteria. The edges with the highest priority are those edges with shortest length and highest connectivity values. The algorithm uses a cutting procedure, labeled network, and sort insertion procedure [4]. The extension of the algorithm consists of the inclusion of a buffer and additional refueling strategies to avoid route failure due to running out of fuel.

The algorithm stores the information of the relevant nodes and edges from the known road network. The cutting procedure removes non-valuable nodes and edges from the known road network. Thus, road edges and road crossings that do not belong to a path from the DMC to at least one victim node are not relevant for the exploration. We use a procedure to cut the unnecessary parts of the road network such as sub-networks completely disconnected from the network and cycles composed of redundant edges. We determine a rooted spanning tree with the DMC as the root to detect the disconnected sub-networks.

The information from the road network at each time $t$ of the exploration is used to assess the accessibility to the victim locations, cut disrupted edges, and evaluate the relevance of nodes and edges for reaching victim locations. The cutting procedure removes non-valuable nodes and edges from the disrupted network at each time $t$ of the exploration.

Cycles in the network are identified through the road network information at time $t$ of the exploration. The road network information is constantly updated, so the on-line algorithm knows after each exploration which edges are still functional in the network and which edges connect which nodes. Thus, the on-line algorithm detects which nodes are reachable from the DMC using a set of functional road edges. In the on-line algorithm, we denote connected nodes and connected edges whenever there is a set of functional and explored edges connecting these nodes and edges to the DMC. Then, connected nodes and connected edges are reachable from the DMC. Finally, the road network information is updated after each cutting procedure. This cutting process is designed to save time by avoiding non-valuable explorations.

### 4.1. Strategy for Dealing with Disrupted Road Networks

After the cutting procedure, the on-line algorithm labels each road edge with its priority to be explored. The sets of edges and priorities define the labeled network. The labeled network is used to guide the UAV to obtain the accessibility information of the victim nodes within the shortest time.

This orientation criterion depends on the connectivity and the length of an edge. The priority value of each edge is the weighted sum of the edge connectivity (*edgeC*) and the edge length (*edgeL*). An importance level $\alpha$ in the co-domain $[0.2, 0.8]$ is given to compute the edge length criterion. Equation (1) computes the edge weight criterion (*edgeW*) for each edge in the disrupted network [4]. The parameter *longest* denotes the length of the longest edge, and the parameter *highestC* is the highest edge connectivity value. Initially, the parameters *edgeL* and *highestC* are determined from the known road network; after the first exploration, these values are calculated using the disrupted network as a reference.

Thus, only the edges relevant for reaching victim locations are those that have an assigned edge weight criterion.

$$edgeW := \alpha \cdot (1 - \frac{edgeL}{longest}) + (1 - \alpha) \cdot \frac{edgeC}{highestC} \tag{1}$$

To apply the orientation criteria, the on-line algorithm uses a sort insertion procedure to determine an efficient UAV routing.

The sort insertion procedure selects the best edge to be explored next, i.e., the edge with the highest priority. The weight criterion is sorted in descending order; thus, the edge with the highest value has the highest priority. In each iteration, the on-line algorithm inserts the best edge into the exploration route of the UAV. Once a road edge is explored, the labeled network has to be updated. The movement strategy is used when:

- A disrupted edge is detected;
- The node that is the current position of the UAV has no adjacent edges to visit.

Taking the edge weight criterion and the current location of the UAV as a reference, the *next-best edge to explore* is selected. Initially, the UAV position is the DMC. If there is no adjacent edge to explore, the back and forward strategy is applied. Otherwise, the adjacent edge with the highest edge weight criterion is selected as the *next-best edge to explore*. Once the *next-best edge to explore* is selected, the UAV explores it by flying over it. Then, the road information of the last exploration is transferred to conduct the cutting procedure and calculate the labeled road network. Thus, the road network information is updated for the following routing decision. This procedure is repeated until the on-line algorithm has the information on the road accessibility of all victim nodes from the DMC.

The back and forward strategy aims to redirect the UAV based on the edges and nodes, reachable from the DMC, detected up to the moment the disruption takes place. An aerial edge or a road edge, or a combination of both, is used by the UAV to return to the closest reachable node with unexplored adjacent edges.

*4.2. Strategy for Dealing with Flying Range Limitations*

Based on [4], we consider a dynamic buffer that ensures that the UAV has enough flying capacity to finish the exploration route at the DMC at any time. The dynamic buffer represents the distance that the UAV has to be able to fly to explore the next selected unexplored road edge *edgeToInsert* and and fly to the closest refueling station. For example, Figure 7 exemplifies how the buffer is calculated, assuming that the DMC is the only reachable refueling station and that the *edgeToInsert* is (4,2), so the aerial edge (2,0) refers to the road edge connecting (4,2) with the reachable refueling station and the next selected unexplored road edge. In Figure 7a, the current position of the UAV is a disruption location, so the buffer is calculated as the sum of the distances of the aerial edge (disruption location, 4), road edge (4,2), and aerial edge (2,0). Similarly, in Figure 7b, the current location of the UAV is a road crossing, whereby the buffer is calculated as the sum of the distance of the road edge (4,2) and aerial edge (2,0).

The buffer depends on the current location (*position*) of the UAV and the location of the closest refueling station. The buffer (*buffer*), the flying range (*flyingRange*), and the flying capacity (*flyingCapacity*) of the UAV are expressed in units of distance. When the UAV is fully fueled, it is assumed that the flying capacity of UAV is the flying range limitation. After, the first exploration, the flying capacity is defined as the difference between the flying range and the travel distance of the UAV since its last refueling operation.

We propose three refueling strategies to deal with the flying range limitation:

- *Extremely conservative refueling strategy (EConservative)*: refuel each time when at the refueling station or refuel when the flying capacity is lower than the buffer needed at that time.
- *Moderately conservative refueling strategy (MConservative)*: refuel when the flying capacity is lower than 50% of the flying range.

- *Conservative refueling strategy (Conservative)*: refuel when the flying capacity is lower than the buffer needed at that time.

The sort insertion procedure applies an additional function to control the flying capacity of the UAV, which refueling stations are available at any given time *t* during the exploration, in turn, to control the buffer depending on the current location of the UAV and available refueling stations. Algorithm 1 presents the extension of the on-line algorithm presented in [4].

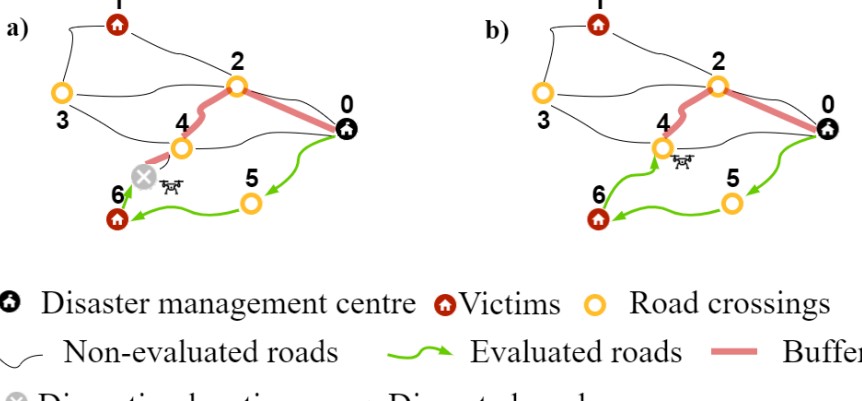

● Disaster management centre  ●Victims  ○  Road crossings

  ⌣ Non-evaluated roads   ⌒ Evaluated roads  ━ Buffer

  ⊗ Disruption locations ⋯ Disrupted roads

**Figure 7.** Example of the dynamic buffer.

---

**Algorithm 1** Control flying capacity procedure.

---

1: **procedure** APPLYINGREFUELINGSTRATEGY(*connectedElements, refuelingStrategy, explorationRoute, flyingRange, position, edgeToInsert, refuelingStations*)
2:    *flyingCapacity* ← computeFlyingCapacity(*flyingRange,explorationRoute*)
3:    *refuelingStation* ←closestStation(*position, refuelingStations, connectedElements*)
4:    *buffer* ← computeBuffer(*position, refuelingStation, edgeToInsert*)
5:    **if** *flyingCapacity* < *buffer* **then**
6:        *explorationRoute* ← addAerialEdge(*position, refuelingStation*)
7:        *flyingCapacity* ←restartFlyingRange(*flyingRange*)                  ▷ **refueling operation**
8:        *explorationRoute* ← recoverRoute(*refuelingStation, edgeToInsert*)
9:        *position* ← updateUAVPosition(*explorationRoute*)
10:    **end if**
11:    **if** *refuelingStrategy* is *EConservative* **then**
12:        **if** *position* ∈ *refuelingStations* & *position* ∈ *connectedElements* **then**
13:            *flyingCapacity* ←restartFlyingRange(*flyingRange*)              ▷ **refueling operation**
14:        **end if**
15:    **end if**
16:    **if** *refuelingStrategy* is *MConservative* **then**
17:        **if** *position* ∈ *refuelingStations* & *position* ∈ *connectedElements* **then**
18:            **if** *flyingCapacity* < 0.5 · *flyingRange* **then**
19:                *flyingCapacity* ←restartFlyingRange(*flyingRange*)          ▷ **refueling operation**
20:            **end if**
21:        **end if**
22:    **end if**
23:    **if** *refuelingStrategy* is *Conservative* **then**
24:        **if** *position* ∈ *refuelingStations* & *position* ∈ *connectedElements* **then**
25:            **if** *flyingCapacity* < *buffer* **then**
26:                *flyingCapacity* ←restartFlyingRange(*flyingRange*)          ▷ **refueling operation**
27:            **end if**
28:        **end if**
29:    **end if**
30:    **return** *explorationRoute*
31: **end procedure**

---

## 5. Computational Experiments

In the following, we present the computational experiments. We first introduce the instances used in Section 5.1, then we discuss the parametrization of the experiments in Section 5.2.

### 5.1. Instance

Taking the instances used in [4] as a reference, we randomly include refueling stations. The duration of the refueling operations includes the time spent on landing and take-off. For the landing and take-off of the UAV, a total time of 10 min is estimated, 5 min for each operation [36,37]. The refueling operation is a parameter that varies depending on the distance traveled by the UAV after its last refueling operation. We assume that the speed of the UAV is 60 km per hour. It is assumed that the landing time, take-off time, and reloading speed are the same for all UAVs. We consider the same UAV types as presented in [4].

Table 2 presents a characterization and clustering of the instances. To cluster the instances, we calculated a relation factor for each instance, and the relation factor for each instance is given by: *number of nodes* times *longest aerial edge* times Total distance of road network (instance) divided by Total distance of road network (largest instance). The relation factor allows us to determine a clustering criterion. In the last column of Table 2, we indicate which instance belongs to which cluster. Instances in clusters A, B, and C are classified as small-, large-, and medium-sized instances, respectively.

**Table 2.** Characteristics of instances and cluster type.

| Instance | Nodes | Total Distance of Road Network (km) | Longest Aerial Edge * (km) | Relation | Cluster |
|---|---|---|---|---|---|
| p1.2.b | 32 | 84,990.26 | 120.07 | 220 | A |
| p2.2.a | 20 | 24,710.51 | 126.84 | 42 | A |
| p3.2.a | 32 | 95,884.38 | 169.03 | 350 | A |
| p4.2.a | 99 | 1,481,730.60 | 257.97 | 25,539 | B |
| p5.2.a | 65 | 351,215.83 | 109.66 | 7128 | C |
| p6.2.a | 63 | 232,134.88 | 130.38 | 5533 | C |
| p7.2.a | 101 | 344,621.25 | 49.92 | 5042 | C |
| * Longest aerial edge connecting to the DMC node | | | | | |

### 5.2. Problem Parametrization

Taking the results of [4] as a reference, the best exploration strategy corresponds to a back and forward movement strategy and a weighted orientation criteria given by $\alpha = \{0.2, 0.3, 0.4, 0.5, 0.6, 0.7, 0.8\}$. The experiments in this section are designed to determine the best refueling strategy in terms of the exploration route duration. The parameterization of the experiments is as follows:

- Ten random seeds.
- Different levels of disruption, given in $\{0.1, 0.2, 0.3, 0.4, 0.5, 0.6, 0.7\}$. Disruptions are classified into three levels, low = $\{0.1, 0.2, 0.3\}$, medium = $\{0.4, 0.5\}$, and high = $\{0.6, 0.7\}$.
- Number of refueling stations available in the disrupted network: (1) only the DMC as refueling station, (2) the DMC and 50% of the victim nodes operate as potential refueling stations, or (3) the DMC and all victim nodes operate as potential refueling stations.
- Three refueling strategies: extremely conservative refueling strategy (*EConservative*), moderately conservative refueling strategy (*MConservative*), and conservative refueling strategy (*Conservative*).

This section presents the following experiments:

- Comparison of the refueling strategies: *EConservative*, *MConservative*, *Conservative*, seven $\alpha$ values, low, medium, and high disruption levels.

- Comparison of the performance of refueling strategies considering in the disrupted network: (1) only the DMC operate as refueling station, (2) the DMC and 50% of the victim locations as potential refueling stations and the DMC, and (3) the DMC and all victim locations as potential refueling stations and the DMC.
- For each disruption level, 10 different disrupted road networks are evaluated per instance (10 seeds).

## 6. Computational Results

In our first numerical study, we are interested in the best strategy. We computed the exploration route for each instance and each seed with a fixed disruption level and $\alpha$ value for all refueling strategies considering only the DMC as refueling station. Then, we determined the best exploration route, i.e., the exploration route with the lowest total travel time, for each instance, each seed, and each $\alpha$ value. This exploration route is compared to all other routes of the same instance, seed, and each $\alpha$ value to determine the best refueling strategy. In the next step, we count over all instance, seeds, and $\alpha$ values how many times each strategy provides the best solution, i.e., the exploration route with the minimum travel time. Strategy ties were added independently to each of the strategies involved. Figure 8 presents the comparison between the strategies. The results indicate that with a percentage around 6%, 49%, and 45%, the best solutions are obtained with a *EConservative*, *MConservative*, and *Conservative* refueling strategy, respectively. Thus, the best refueling strategy is *MConservative*, followed by *Conservative*, and the worst refueling strategy is *EConservative*.

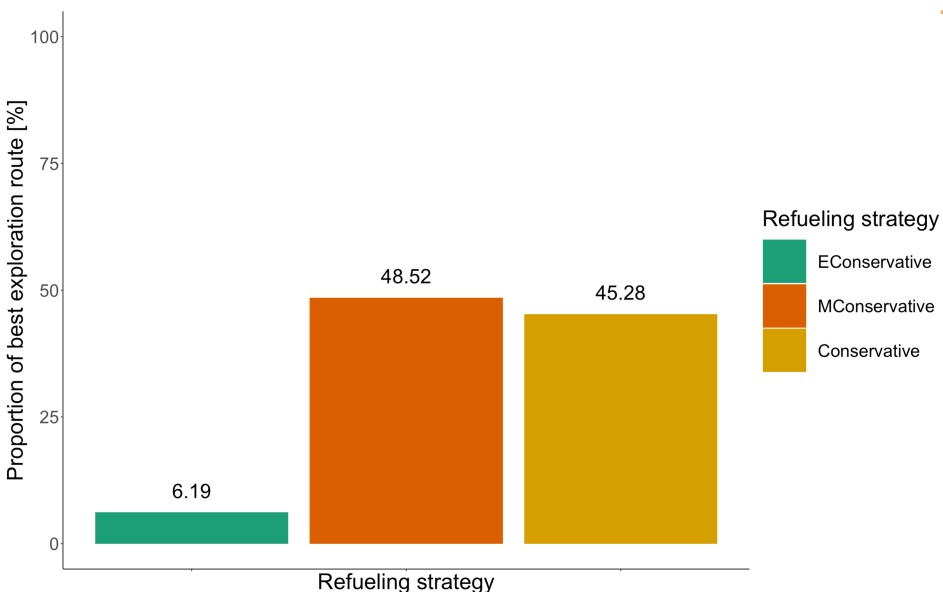

**Figure 8.** Global performance of refueling strategies.

In the second stage of the numerical study, we computed the average duration of the exploration route for each instance and each disruption level for all $\alpha$ values and refueling strategies. Then, we determined the average duration of the exploration route for each instance and each disruption level. In the next step, for each instance and each refueling strategy, we clustered the average duration of the exploration route by low, medium, and high disruption level. Then, these values were taken to compare average values over all instances. Figures 9–11 present for each refueling strategy the average duration of the exploration route for each instance and disruption level cluster.

The moderately conservative refueling strategy *MConservative* reports the best route exploration. This is especially clear for situations where the road network has a disruption level of up to 50%. For disruption levels of 50% and more, the performance of all strategies is similar. The similar performance is due to the longer time it takes for the UAV to find a functional edge connecting the DMC to the road network when disruption levels are higher, so finding a refueling station in addition to the DMC also takes longer. Therefore, many refueling operations are performed at the DMC. Depending on the characteristics of the road network, the number of refueling operations required varies slightly between refueling strategies.

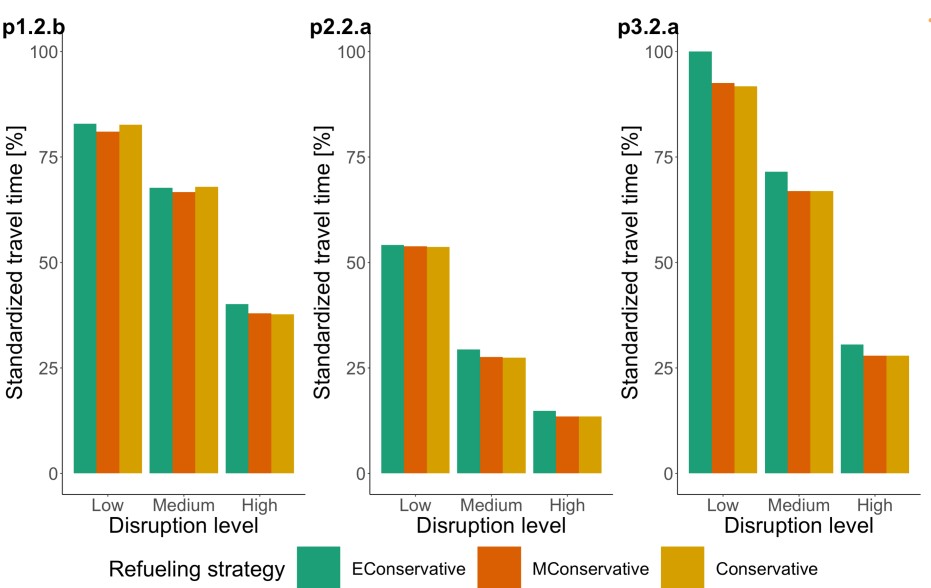

**Figure 9.** Performance of refueling strategies: normalized travel time of exploration route for instance cluster A. The standardized travel time 100% is equivalent to 21.56 h.

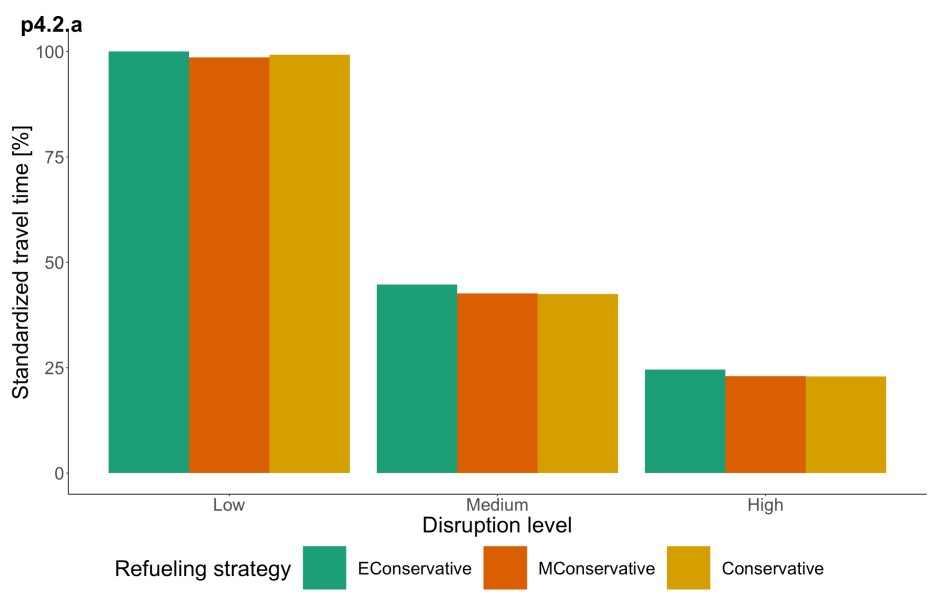

**Figure 10.** Performance of refueling strategies: normalized travel time of exploration route for instance cluster B. The standardized travel time 100% is equivalent to 57.52 h.

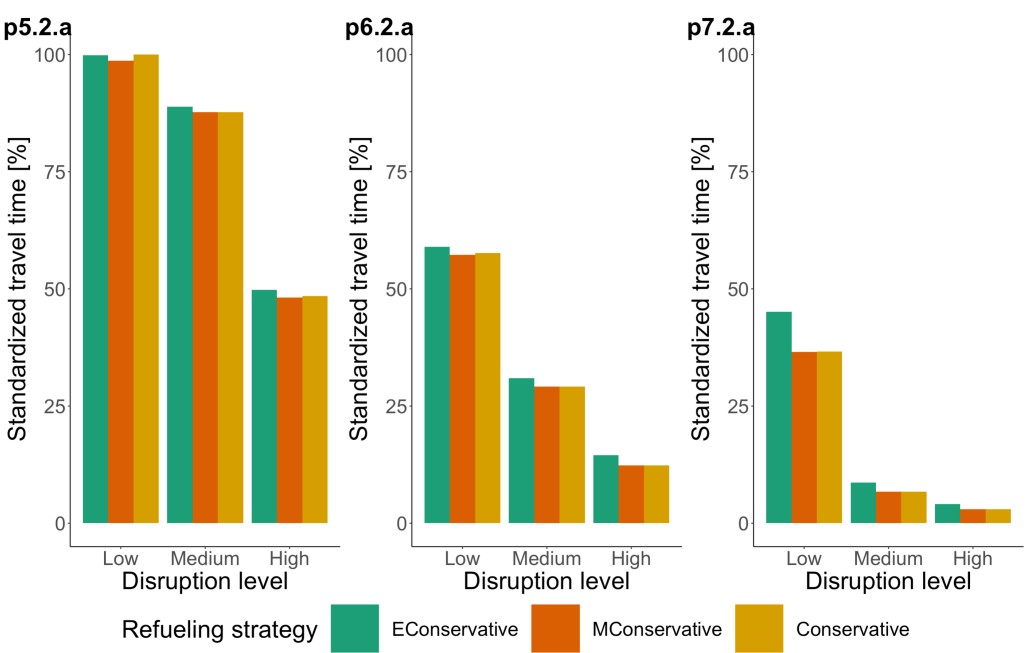

**Figure 11.** Performance of refueling strategies: normalized travel time of exploration route for instance cluster C. The standardized travel time 100% is equivalent to 38.23 h.

The last numerical study is to determine the impact of having multiple refueling stations: (1) only the DMC as refueling stations, (2) the DMC and 50% of the victim nodes operate as potential refueling stations, or (3) the DMC and all victim nodes operate as potential refueling stations. Remember that the victim locations operate as refueling stations only if they are known to be reachable from the DMC using the road network. We computed the average duration of the exploration route for each instance and each disruption level for all $\alpha$ values and number of refueling stations. Then, we determined the average duration of the exploration route for each instance and each disruption level. In the next step, for each instance and refueling strategy, we clustered the average duration of the exploration route by low, medium, and high disruption levels. Then, these values were taken to compare average values over all instances. Figures 12–14 present for each number of refueling stations the average travel time savings of the UAV for each refueling strategy, disruption level, and instance cluster. We computed the worst exploration route over all $\alpha$ values for each instance, seed, disruption level, and refueling strategy, i.e., the time-longest exploration route. The worst solution of each instance, seed, disruption level, and refueling strategy is compared to the other solutions of the same instance, disruption level, refueling strategy, and number of refueling stations. In the next step, we calculated the difference between the total travel time of the worst solution and the total travel time of the other solutions. The difference is calculated for each refueling strategy; the difference is conceived of as a savings in travel times that can be achieved depending on the number of refueling stations in the disrupted network. Considering the clusters of instances in Table 2, we calculated the average savings for each level of disruption, refueling strategy, and number of refueling stations over all instances, seed, and $\alpha$ values.

The following is the analysis of the results for each refueling strategy. The results show that for road networks with a low, medium, and high level of disruption, using the refueling strategy *EConservative* (Figures 12–14, *EConservative*), the worst exploration route is achieved when considering 100% of the victim locations as refueling stations. These results are argued on the fact that there is a high chance that victim nodes are accessible in networks with a low level of disruption. Therefore, every time a disruption is detected, the exploration route is redirected to the last accessible explored node in the disrupted road network, being a victim node or road crossing. Thus, each route redirection can lead to a refueling operation, increasing the exploration duration. Thus, the results show

that for instances in cluster A and C with a low disruption level using refueling strategy *EConservative*, the best solution is achieved when only the DMC operates as refueling station. Similar for instances in cluster A and B with a medium disruption level, the best solution is achieved when only the DMC operates as a refueling station.

The results for networks with a high disruption level using the *EConservative* refueling strategy differ among the clustering of instances. The best solutions for instances in cluster A are obtained when 50% of the victim locations operate as potential refueling stations. For cluster B, the best solutions are also achieved when 50% of the victim locations operate as potential refueling stations. For instances in cluster C with high disruption levels, the best solution is achieved when only the DMC operates as a refueling station.

Similarly, we analyzed the results of the *MConservative* and *Conservative* refueling strategies (Figures 12–14, *MConservative* and *Conservative*). The worst solution occurs when only the DMC is considered as the refueling station for the *MConservative* and *Conservative* refueling strategies. This is because every time a refueling operation is needed, the UAV has to fly back to the DMC. The best solution for the *EConservative* and *Conservative* refueling strategies is achieved when 100% of the victim locations are considered as refueling stations.

In conclusion, the refueling strategy *MConservative* is the best strategy to deal with flying range limitations for small and large instance clusters with a low disruption. For small and large instance clusters with a high disruption level, the best refueling strategy is the strategy *Conservative*. For medium-sized instances, the best refueling strategy varies between strategy *MConservative* and strategy *Conservative* depending on the disruption level of the road network and the number of refueling stations available. Furthermore, for all instance clusters with a medium disruption level, the best refueling strategy can be strategy *MConservative* or strategy *Conservative* depending on the number of refueling stations available. The refueling strategies *MConservative* and *Conservative* have a better balance between returning to a refueling station due to the need for a refueling operation (the flying capacity is less than the buffer) and the cost of advancing on the exploration route until the UAV visits a refueling station by coincidence.

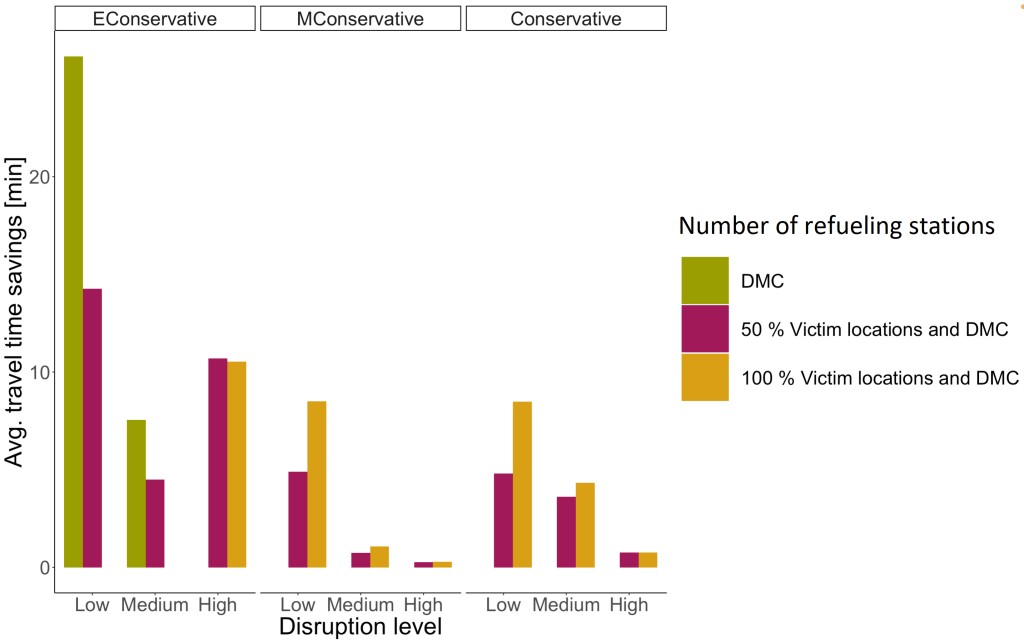

**Figure 12.** Cluster A: impact of the number of refueling stations on the performance of refueling strategies.

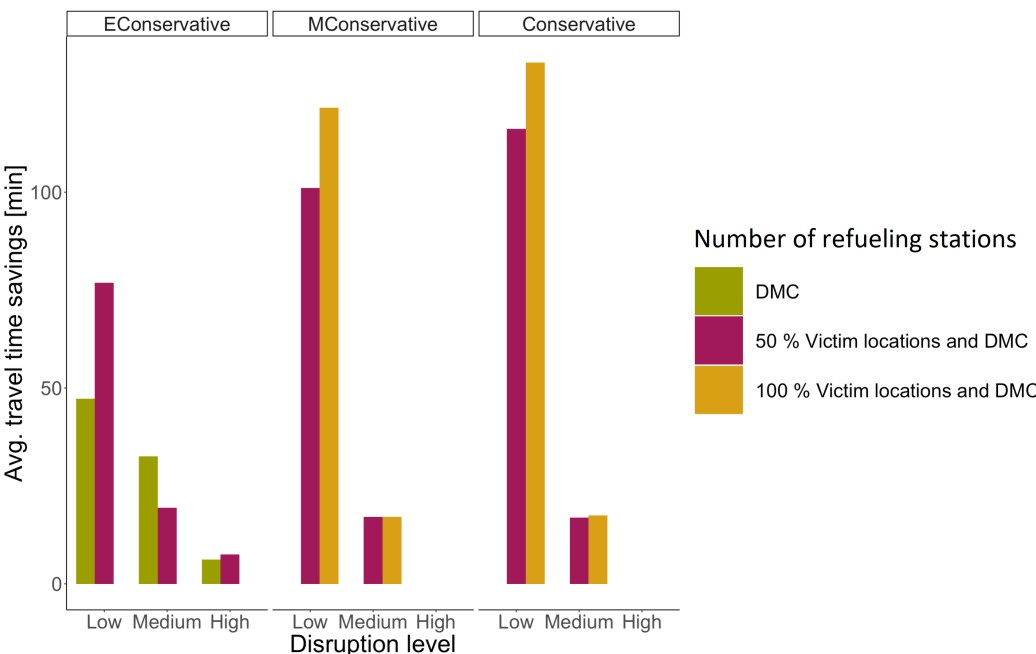

**Figure 13.** Cluster B: impact of the number of refueling stations on the performance of refueling strategies.

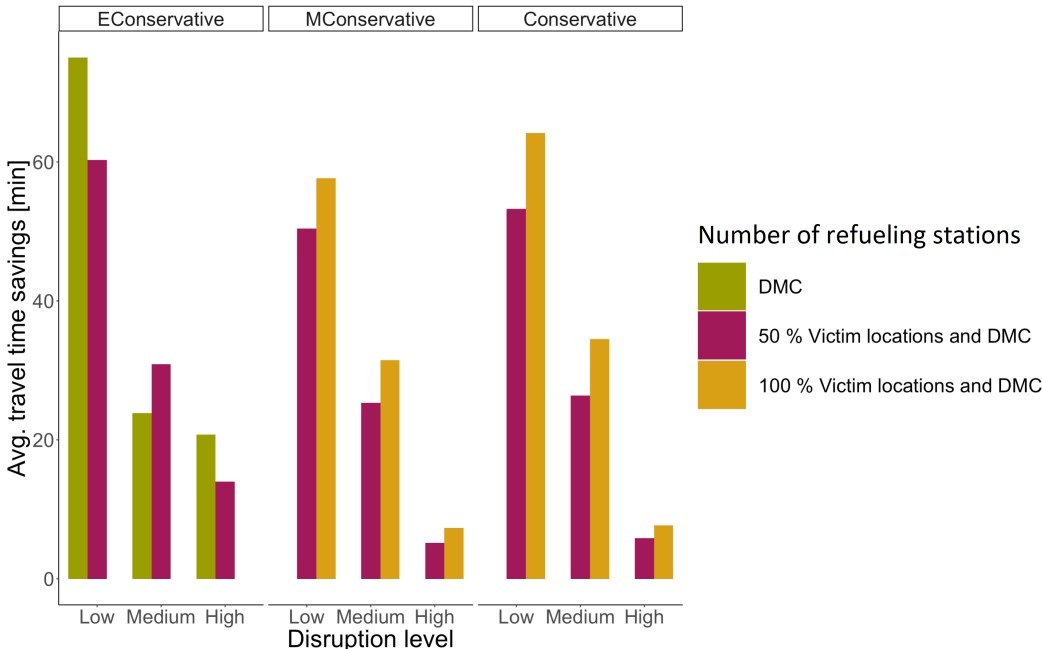

**Figure 14.** Cluster C: impact of the number of refueling stations on the performance of refueling strategies.

The results regarding the best refueling strategy varies depending on the the number of refueling stations, instance size, and disruption level. Table 3 presents a consolidation of the results; the first two columns refer to the disaster situation, and the last two columns present the best refueling strategy to reach the exploration routes with the lowest travel time given the number of refueling stations in advance. We determined the best refueling strategy for each instance cluster, each disruption level, and each number of refueling stations.

**Table 3.** Consolidation of the best combination of refueling strategies and number of refueling stations available.

| Disaster Situation | | | Best Strategy | | |
|---|---|---|---|---|---|
| Instance Size | Disruption Level | Number of Refueling Stations | *EConservative* | *MConservative* | *Conservative* |
| Small | low | DMC | | x | |
| | | 50% of victim locations | | x | |
| | | 100% of victim locations | | x | |
| | Medium | DMC | | x | |
| | | 50% of victim locations | | x | |
| | | 100% of victim locations | | x | |
| | High | DMC | | | x |
| | | 50% of victim locations | | | x |
| | | 100% of victim locations | | | x |
| Medium | low | DMC | | x | |
| | | 50% of victim locations | | x | |
| | | 100% of victim locations | | | x |
| | Medium | DMC | | x | |
| | | 50% of victim locations | | x | |
| | | 100% of victim locations | | | x |
| | High | DMC | | x | |
| | | 50% of victim locations | | x | |
| | | 100% of victim locations | | x | |
| Large | low | DMC | | x | |
| | | 50% of victim locations | | x | |
| | | 100% of victim locations | | x | |
| | Medium | DMC | | | x |
| | | 50% of victim locations | | | x |
| | | 100% of victim locations | | | x |
| | High | DMC | | | x |
| | | 50% of victim locations | | | x |
| | | 100% of victim locations | | | x |

## 7. Conclusions

This paper investigated the exploration route of a UAV to explore a disrupted road network, i.e., a road network after a disaster considering flying range limitations. The study is based on a conservative exploration strategy called back and forward. After each detected disruption, the UAV is redirected to the closest node connected to the functional road network detected so far. We proposed three refueling strategies to deal with the flying range limitation. The three refueling strategies were tested using a weight orientation criterion. This orientation criterion aims to find the combination of strategies that minimizes the duration of the exploration route to determine whether or not the given victim locations can be reached via the road network.

The refueling strategies were proposed to extend the coverage of the UAV flying range. The algorithm was designed to save travel time by avoiding unnecessary network explorations. We assumed that a certain portion of victim locations can operate as refueling stations if they are connected by road to the DMC. This assumption is realistic as once a victim location is found to be reachable, it is certain that a functional path from the DMC to this victim location exists, and thus, the victim node can be resupplied with additional fuel (for the refueling station). The results show that the refueling strategy *MConservative*, where the refueling operation takes place when the flying capacity is lower than 50%, provides the best exploration routes for all instance clusters with a low disruption level.

The refueling strategy *Conservative*, where the refueling operation is performed only when the flying capacity is equal to or less than the buffer, provides the best solution for small and large instance clusters with a high level of disruption. The best strategy for all instances with a medium disruption level varies, depending on the number of refueling stations available, between *MConservative* and *Conservative*. The most significant impact on total exploration time, apart from network size and disruption level, which cannot be influenced, had the fuel buffer and the number of reachable refueling stations. Therefore, decision-makers must establish criteria to decide when to refuel and where the refueling should take place before a disaster hits.

Future research lines of this work can be established for reconstruction operations. We are interested in studying the exploration of the disrupted network to make decisions on road recovery to improve accessibility to the affected area. In addition, loading capacity constraints can be considered to study the distribution of humanitarian kits while the UAV explores the state of the disrupted network. Our solution approach can be extended by considering a periodical exploration to capture the dynamism of the catastrophe.

**Author Contributions:** Conceptualization, L.R.-R., J.V., and P.H.; methodology, L.R.-R., J.V., and P.H.; software, L.R.-R.; validation, L.R.-R., J.V., and P.H.; formal analysis, L.R.-R., J.V., and P.H.; investigation, L.R.-R.; resources, L.R.-R., J.V., and P.H.; data curation, L.R.-R.; writing—original draft preparation, L.R.-R.; writing—review and editing, L.R.-R., J.V., and P.H.; visualization, L.R.-R.; supervision, P.H. All authors have read and agreed to the published version of the manuscript.

**Funding:** This research received no external funding.

**Institutional Review Board Statement:** Not applicable.

**Informed Consent Statement:** Not applicable.

**Data Availability Statement:** The files with the test instances and their detailed parameterization of the on-line algorithm are available at short.boku.ac.at/instances (accessed on 17 June 2022).

**Acknowledgments:** We would like to thank Sven Müller for his support in the development of the paper. Further, we would like to thank Klaus-Dieter Rest from the University of Natural Resources and Life Sciences Vienna and Javier Faulin from the Public University of Navarra for their valuable insights in the first phase of this research.

**Conflicts of Interest:** The authors declare no conflict of interest.

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
