# Peer review of "An Online Algorithm for Routing an Unmanned Aerial Vehicle for Road Network Exploration Operations after Disasters under Different Refueling Strategies"

_algorithms, doi:10.3390/a15060217_

Round 1
Reviewer 1 Report
Please, check references (e.g., reference number 7).
Improve the quality of figure 1.
Please, clarify the position of the paper in current literature and make explicit the advancements respect to your previous work.
Literature review: I suggest improving and update the structure of this section, by introducing a (brief) sub-section related to humanitarian logistics and a (brief) sub-section on network disruption. Some suggestions:
· Prepositioning and distributing relief items in humanitarian logistics with uncertain parameters, Elsevier
· Points of distribution location and inventory management model for Post-Disaster Humanitarian Logistics, Transportation Research Part E, Elsevier
· A scenario-based possibilistic-stochastic programming approach to address resilient humanitarian logistics considering travel time and resilience levels of facilities, International Journal of Systems Science: Operations & Logistics, Taylor and Francis
· Impact of transport network disruption on travel demand: A case study of the July 2018 heavy rain disaster in Japan, Asian Transport Studies, Elsevier
· Emergency Response after Disaster Strikes: Agent-Based Simulation of Ambulances in New Windsor, NY. Journal of Infrastructure Systems
· Fleet routing and scheduling in bushfire emergency evacuation: A regional case study of the Black Saturday bushfires in Australia, Transportation Research Part D
· …
Section 3, line 135: DMS is included in the set N? Please, clarify.
Section 3: please, can you better explain the relation among N, N^t, Nt and among E, E^t, At?
Section 3: assumption 1, please justify the choice to consider undirected the road network; clarify assumption 11 (why 50% or 100%); assumption 14 is useless.
Note that 10/15 assumption are from a previous work. The novelty seems be the fuelling station position: if a victim location is a fuelling station, this location must be reached by a land vehicle able to refuel the UAV. Is that so? And this case, how the route on the road network is congruent with the direction of the edges (see assumption 1)? Please, clarify this point.
Check table 1, last column is not readable.
The computational part is limited to low dimension problems. What would happen in a full-sized network? An example of full-sized networks: https://www.bgu.ac.il/~bargera/tntp/.
Author Response
Dear Reviewers and dear Editor,
We would like to thank the Reviewers and the Area Editor for the time they have invested in reviewing our work, and for the useful feedback they provided.
The answers to your comments are in the attached file.
The authors

Reviewer 2 Report
See attached review.

Author Response

(The authors gave the same response as above.)

Round 2
Reviewer 2 Report
See attached,
